# Comparison of sonication with chemical biofilm dislodgement methods using chelating and reducing agents: Implications for the microbiological diagnosis of implant associated infection

Svetlana Karbysheva[1,2], Mariagrazia Di Luca[1,2], Maria Eugenia Butini[1,2], Tobias Winkler[1,2], Michael Schütz[3], Andrej Trampuz[1,2]*

**1** Center for Musculoskeletal Surgery, Charité –Universitätsmedizin Berlin, corporate member of Freie Universität Berlin, Humboldt-Universität zu Berlin and Berlin Institute of Health, Berlin, Germany, **2** Berlin-Brandenburg Centre for Regenerative Therapies (BCRT), Berlin, Germany, **3** Department of Orthopaedics and Trauma, Jamieson Trauma Institute, Queensland University of Technology, Brisbane, Australia

* andrej.trampuz@charite.de

**Data Availability Statement:** All relevant data are within the manuscript.

## Abstract

The diagnosis of implant-associated infections is hampered due to microbial adherence and biofilm formation on the implant surface. Sonication of explanted devices was shown to improve the microbiological diagnosis by physical removal of biofilms. Recently, chemical agents have been investigated for biofilm dislodgement such as the chelating agent ethylenediaminetetraacetic acid (EDTA) and the reducing agent dithiothreitol (DTT). We compared the activity of chemical methods for biofilm dislodgement to sonication in an established *in vitro* model of artificial biofilm. Biofilm-producing laboratory strains of *Staphylococcus epidermidis* (ATCC 35984), *S. aureus* (ATCC 43300), *E. coli* (ATCC 25922) and *Pseudomonas aeruginosa* (ATCC 53278) were used. After 3 days of biofilm formation, porous glass beads were exposed to control (0.9% NaCl), sonication or chemical agents. Quantitative and qualitative biofilm analyses were performed by colony counting, isothermal microcalorimetry and scanning electron microscopy. Recovered colony counts after treatment with EDTA and DTT were similar to those after exposure to 0.9% NaCl for biofilms of *S. epidermidis* (6.3 and 6.1 vs. 6.0 $\log_{10}$ CFU/mL, *S. aureus* (6.4 and 6.3 vs. 6.3 $\log_{10}$ CFU/mL), *E. coli* (5.2 and 5.1 vs. 5.1 $\log_{10}$ CFU/mL and *P. aeruginosa* (5.1 and 5.2 vs. 5.0 $\log_{10}$ CFU/mL, respectively). In contrast, with sonication higher CFU counts were detected with all tested microorganisms (7.5, 7.3, 6.2 and 6.5 $\log_{10}$ CFU/mL, respectively) ($p < 0.05$). Concordant results were observed with isothermal microcalorimetry and scanning electron microscopy. In conclusion, sonication is superior to both tested chemical methods (EDTA and DTT) for dislodgement of *S. epidermidis*, *S. aureus*, *E. coli* and *P. aeruginosa* biofilms. Future studies may evaluate potential additive effect of chemical dislodgement to sonication.

**Funding:** This work was funded by PRO-IMPLANT Foundation (https://www.pro-implant-foundation.org), a non-profit organization supporting research, education, global networking and care of patients with bone, joint or implant-associated infection. The funding had no influence on the data analysis or interpretation of the results.

**Competing interests:** The authors have declared that no competing interests exist.

## Introduction

Implants are increasingly used to improve the mobility (joint replacement and bone fixation devices) or prolong the survival and assist the performance of physiological functions (cardiac implantable electronic device (CIED) and neurosurgical shunts). Infections represent a significant complication of implant surgery, resulting in major challenges regarding the diagnosis and treatment [1–5]. Most commonly isolated microorganisms in patients with periprosthetic joint infection are coagulase-negative staphylococci (30–45%) and *Staphylococcus aureus* (12–23%), followed by streptococci (9–10%), enterococci (3–7%), gram-negative bacilli (3–6%) and anaerobes (2–4%) [6]. Similar distribution of pathogens is observed in CIED [2] and neurosurgical shunt-associated infections [4].

The crucial step in the management of implant-associated infections is an accurate diagnosis. However, as these infections are caused by microorganisms embedded in a polymeric matrix attached to the device surface, the diagnosis may be challenging, especially in chronic low-grade infections. In order to detect the infecting microorganism, dislodgement of the biofilm should precede the standard cultivation methods in solid or liquid growth media [7].

Various approaches had been investigated for biofilm removal from implant surface. Sonication is based on mechanical biofilm dislodgement and showed superior detection yields than other methods and was introduced in routine microbiological diagnosis [8–12].

In vitro studies investigated the ability of chemical dislodgement such as metal-chelating agent ethylenediaminetetraacetic acid (EDTA) and the strong reducing agent dithiothreitol (DTT). The ability of EDTA to chelate and potentiate the cell walls of bacteria and destabilize biofilms by sequestering calcium, magnesium, zinc, and iron suggests to be suitable for the biofilm detachment [13]. Recent reports suggested that treatment of explanted prostheses with a solution containing DTT is superior to sonication for dislodgement of biofilm-embedded bacteria [14].

The aim of the study was to compare the ability of mechanical biofilm dislodgement (i.e. sonication) with chemical dislodgement methods (i.e. EDTA and DTT) *in vitro* and evaluate their potential role in the routine microbiological diagnosis of implant-associated infections.

## Materials and methods

### Bacterial strains and biofilm growth conditions

As a model to form the bacterial biofilm porous glass beads (diameter 4 mm, pore sizes 60 μm, ROBU®, Hattert, Germany) were used. Due to the high volume-to-surface ratio, glass beads were used for biofilm studies rather than smooth materials, as investigated in numerous previous research works regarding biofilm formation and anti-biofilm activity [15–20]. To form biofilms, beads were placed in 2 ml of brain heart infusion broth (BHIb, Sigma-Aldrich, St. Louis, MO, USA) containing $1 \times 10^8$ CFU/mL inoculum of *Staphylococcus epidermidis* (ATCC 35984), *S. aureus* (ATCC 43300) *E. coli* (ATCC 25922) or *Pseudomonas aeruginosa* (ATCC 53278) and incubated at 37˚C. After 24 h, beads were re-incubated in fresh BHIb and biofilms were statically grown for further 72 h at 37˚C, as previously described [14]. After biofilm formation, beads were washed six times with 2 ml 0.9% NaCl to remove planktonic bacteria.

### Biofilm dislodgement by chemical methods (EDTA or DTT) or sonication

To define the minimal chemical concentration and treatment duration for biofilm dislodging, washed beads were placed in 1 ml of EDTA at concentrations 12, 25 and 50 mM or DTT at concentrations 0.5, 1 and 5 g/L and exposed for 5, 15 and 30 min. Untreated beads incubated

with 0.9% NaCl were used as negative control. The timing of EDTA and DTT exposure and choice of the concentration for biofilm dislodgement were based on previous studies, which indicated the maximal biofilm disruption without bacterial killing [13, 14].

To evaluate the sonication effect, biofilms were sonicated as described previously [10]. Briefly, each bead was inoculated in 1 ml 0.9% NaCl, vortexed for 30 sec, sonicated at 40 kHz at intensity 0.1 Watt/cm$^2$ (BactoSonic, BANDELIN electronic, Berlin, Germany) for 1 min and vortexed again for 30 sec. One-hundred microliter of serial dilutions of the resulting sonication fluid or the solution obtained after chemical treatment with DTT or EDTA were plated onto Tryptic Soy Agar (TSA) (Sigma-Aldrich, St. Louis, MO, USA). After 24 h of incubation at 37˚C, the CFU/mL number was counted. The serial dilutions allowed to raise the upper limit of detection providing a reportable range from 0 to 100,000,000 CFU/mL.

Additionally, the viability of planktonic bacteria in presence of chemical agents and sonication was evaluated. Planktonic cells of *S. epidermidis*, *S. aureus*, *E. coli and P. aeruginosa* at final concentration of $\approx 10^5$ CFU/ml were exposed to EDTA (25 mM) and DTT (1 g/L) for different time periods (5, 15 and 30 min) and sonication. All experiments were performed in triplicates S1 Fig.

## Isothermal microcalorimetry analysis

To prove the dislodgement effect of previously described methods and reveal the presence of bacterial cells remained attached on the bead surface, treated beads were washed six times in 2 ml 0.9% NaCl to remove the dislodged biofilm and placed in 4 ml-glass ampules containing 3 ml of BHIb. The ampoules were air-tightly sealed and introduced into the microcalorimeter (TAM III, TA Instruments, Newcastle, DE, USA), first in the equilibration position for 15 min to reach 37˚C and avoid heat disturbance in the measuring position. Heat flow (µW) was recorded up to 20 h. The calorimetric time to detection (TTD) was defined as the time from insertion of the ampoule into the calorimeter until the exponentially rising heat flow signal exceeded 100 µW to distinguish microbial heat production from the thermal background [21]. Growth media without bacteria served as negative control.

## Scanning electron microscopy (SEM)

Beads with biofilm were fixed with 2.5% (v/v) glutaraldehyde in natrium cacodylat buffer and the samples were dehydrated with increasing concentrations of ethanol for 2 min each. The samples were stored in vacuum until use. Prior to analysis by Scanning electron microscope (GeminiSEM 300, Carl Zeiss, OberkochenDSM 982 GEMINI, Zeiss Oberkochen, Germany), the samples were subjected to gold sputtering (Sputter coater MED 020, Balzer, BingenMED 020, BAL-TEC). All experiments were performed in triplicate.

## Statistical methods

Statistical analyses were performed using SigmaPlot (version 13.0; Systat Software, Chicago, IL, USA) and graphics using Prism (version 8; GraphPad, La Jolla, CA, USA). Quantitative data were presented as mean ± standard deviation (SD) or median and range, as appropriate. To compare different groups the ANOVA test was performed and in case of significant differences, nonparametric Kruskal-Wallis test and Wilcoxon signed-rank test for independent samples were used, as appropriate. The significance level in hypothesis testing was predetermined at p-value of <0.05.

## Results and discussion

### CFU counting method

**Bacterial biofilm dislodged after treatment with different concentrations of chemical agents at different time points.** *S. epidermidis biofilm*. EDTA at concentration 25 mM showed significant increase in bacterial count at 15 min compared to 5 min (p = 0.023) and compared to 30 min of exposure (p = 0.012). DTT showed no difference in CFU count when different concentrations were applied (Fig 1A and 1B).

*S. aureus biofilm*. EDTA at concentration 25 mM and DTT at concentration 1 g/L at different time points (5, 15 and 30 min) showed no significant difference in CFU count. (Fig 1C and 1D).

*E. coli biofilm*. EDTA at concentration 25 mM and DTT at concentration 1 g/L at diferent time points (5, 15 and 30 min) showed no significant difference in CFU count. (Fig 1E and 1F).

*P. aeruginosa biofilm*. EDTA at concentration 25 mM showed significant increase in bacterial count at 15 min (p = 0.042) and 30 min (p = 0.020) compared to 5 min of exposure. DTT at concentration 1 g/L showed increase in bacterial count at 15 min (p = 0.018) compared to 30 min, and a significant increase in CFU at concentration 5 g/L at 15 min compared to 5 min (p = 0.028) was observed (Fig 1G and 1H).

Therefore to evaluate the dislodgement effect of chemical methods the concentrations 25 mM EDTA and 1 g/L DTT were chosen as they showed significant increase in CFU count at 15 min compared to other time points when *P. aeruginosa* and *S. epidermidis* biofilms were investigated. The mean colony count obtained after treatment of *S. epidermidis* biofilms with EDTA (25 mM, 15 min) and DTT (1 g/L, 15 min) was similar to those observed after treatment with 0.9% NaCl used as control (6.3, 6.1 and 6.0 $\log_{10}$ CFU/mL, respectively). In contrast, sonication detected significantly higher CFU counts with 7.5 $\log_{10}$ CFU/mL (p <0.05) (Fig 2A). Similar results were observed when *S. aureus* biofilms were treated with chemicals (EDTA, 25 mM, 15 min) and DTT, 1 g/L, 15 min) or 0.9% NaCl (6.4, 6.3 and 6.3 $\log_{10}$ CFU/mL, respectively). By using sonication, CFU count of 7.3 $\log_{10}$ CFU/mL (p < 0.05) was observed (Fig 2B).

We found similar colony counts when *E. coli* biofilms were treated with EDTA (25 mM, 15 min) and DTT (1 g/L, 15 min) as well as 0.9% NaCl (5.2, 5.1 and 5.1 $\log_{10}$ CFU/mL, respectively). Sonication detected significantly higher CFU counts with 6.2 $\log_{10}$ CFU/mL (p < 0.05) (Fig 2C). The results were similar when *P. aeruginosa* biofilms were investigated. Treatment with chemicals (EDTA, 25 mM, 15 min) and DTT, 1 g/L, 15 min) or 0.9% NaCl (5.1, 5.2 and 5.0 $\log_{10}$ CFU/mL, respectively). Sonication showed significantly higher CFU counts with 6.5 $\log_{10}$ CFU/mL (p < 0.05), (Fig 2D).

### Isothermal microcalorimetry

Heat produced by samples containing sonicated glass beads with *S. epidermidis* biofilm was detected after 11 h. In contrast, heat production exceeding the threshold of 100 μW was observed earlier (after 6.5 and 6.4 h) for the samples that were previously treated with EDTA and DTT, confirming the presence of a higher number of residual bacteria on beads treated with chemical methods, in comparison to those after sonication. This time difference was statistically significant (p <0.05). No difference in heat production was observed after treatment with 0.9% NaCl (control) and EDTA or DTT (6.3 vs 6.5 and 6.4 h, respectively) (p = 0.3) (Fig 3A). Similar results were observed with the analysis of *S. aureus* biofilm beads. The time of heat detection after sonication of beads was significantly higher (12 h) in comparison to EDTA and DTT (6.1 and 5.8 h, respectively) (p <0.05); no difference between both chemical methods and the control (4.6 h) was observed (Fig 3B). Investigation of *E. coli* and *P. aeruginosa*

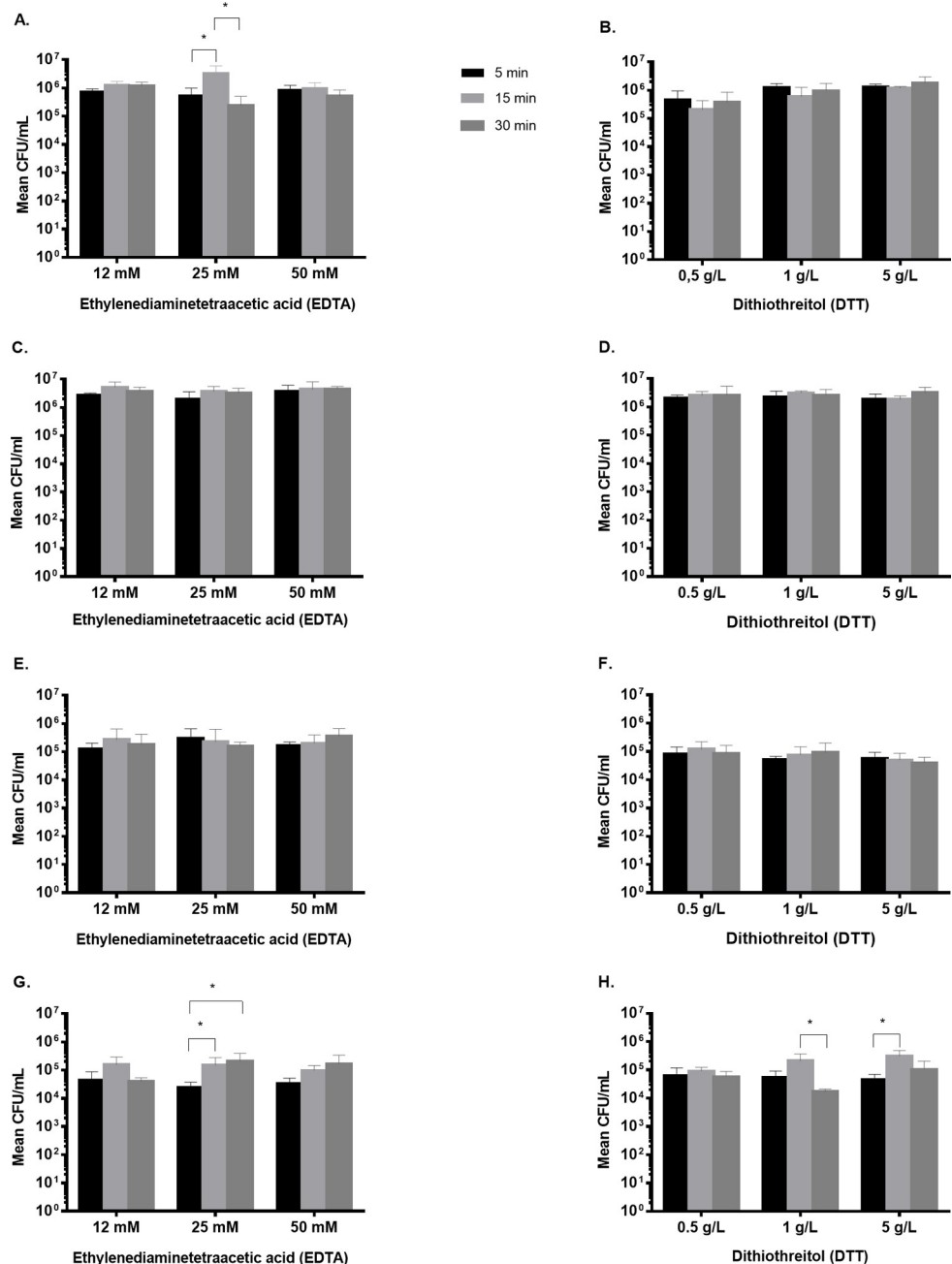

**Fig 1. Bacterial biofilm dislodged after treatment with different concentrations of chemical agents at different time points.** *S. epidermidis* biofilm (A) EDTA, (B) DTT. *S. aureus* biofilm (C) EDTA, (D) DTT; *E. coli* biofilm (E) EDTA, (F) DTT; *P. aeruginosa* biofilm (G) EDTA, (H) DTT. Mean values are shown, error bars represent standard deviation. * Statistically significant difference (p < 0.05).

biofilms showed the same results. Time of heat detection in sonicated beads was significantly higher compared to beads treated with chemical agents (EDTA, DTT) as well as control: 7.8 h vs. 4.9, 4.5 and 4.5 h, respectively (p <0.05) for *E. coli* biofilm and 11h vs. 6.5, 6.5 and 4.6 h, respectively (p <0.05) for *P. aeruginosa* biofilm, (Fig 3C and 3D).

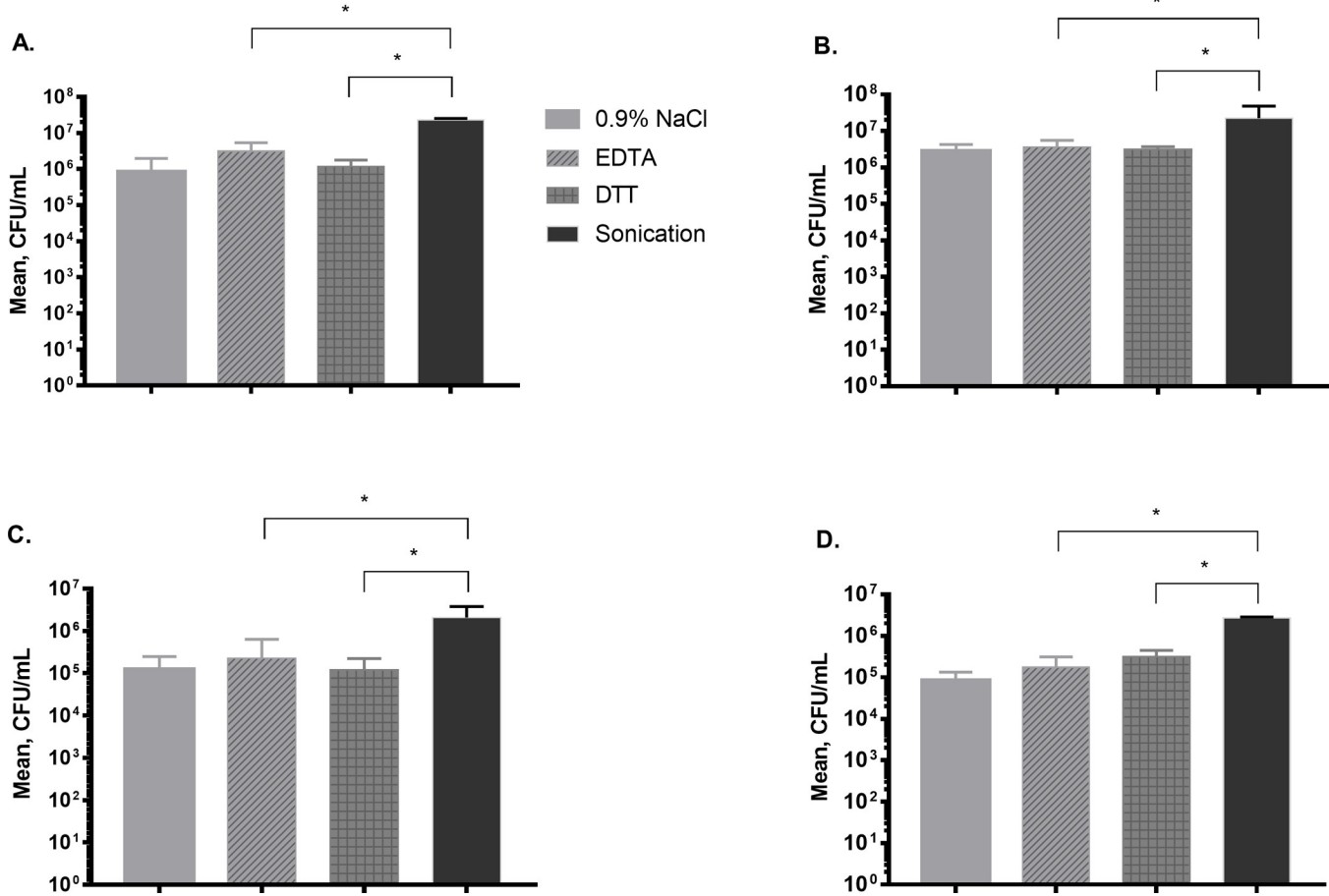

**Fig 2. Quantitative analysis and comparison of biofilm dislodging methods.** (A) *S. epidermidis* biofilm. (B) *S. aureus* biofilm. (C) *E. coli* biofilm. (D) *P. aeruginosa* biofilm. Mean values are shown, error bars represent standard deviation. * Statistically significant difference (p < 0.05). 0.9% NaCl represents an untreated control. EDTA, ethylenediaminetetraacetic acid. DTT, dithiothreitol.

## Scanning electron microscopy

The use of scanning electron microscopy (SEM) allowed to visualize the biofilms of *S. epidermidis*, *S. aureus*, *E. coli* and *P. aeruginosa* before and after treatments with either chemicals or sonication. For all microorganisms the scanning electron microscope images showed substantial less biofilm biomass remaining on the beads when sonication was applied compared to control as well as both chemical methods (Figs 4–7).

Implant-associated infections represent a major challenge for the microbiological diagnosis due to biofilm formation [22, 23]. We investigated the ability of different *in vitro* biofilm dislodgement methods, including sonication as the standard procedure in our institution and chemical treatment using EDTA or DTT as investigational procedures.

For biofilm formation we used laboratory strains of *S. epidermidis*, *S. aureus*, *E. coli* and *P. aeruginosa* known to be good biofilm formers. We did not use clinical strains as they typically show larger variability and are not suitable for investigation of a new diagnostic method. *S. epidermidis* and *S. aureus* were chosen as they are the most common pathogens causing implant-associated infections. *P. aeruginosa* and *E. coli* were chosen as representative pathogens of gram-negative bacteria causing about 10–15% of periprosthetic joint infections, up to 40% of

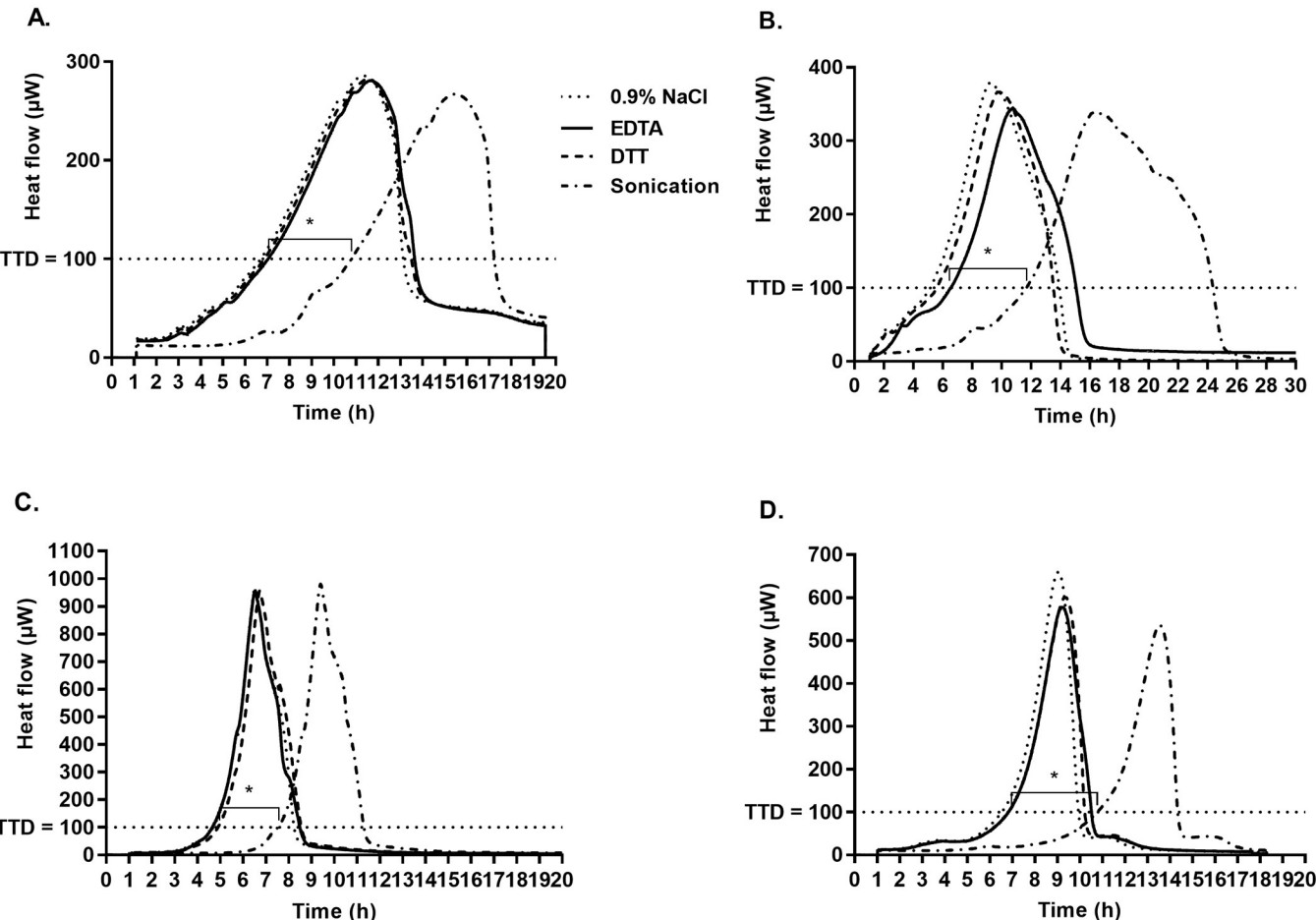

**Fig 3. The microcalorimetric time to detection (TTD) of bacterial growth.** (A) *S. epidermidis* biofilm. (B) *S. aureus* biofilm. (C) *E. coli* biofilm. (D) *P. aeruginosa* biofilm. 0.9% NaCl represents an untreated control. EDTA, ethylenediaminetetraacetic acid. DTT, dithiothreitol.TTD, the calorimetric time to detection of microbial heat production. * Statistically significant difference (p < 0.05).

fracture-fixation device-associated infections and up to 15% of neurosurgical shunt-associated infections [2–5]. The chosen *P. aeruginosa* strain was shown to be a good biofilm producer in previous biofilm studies [24].

To compare the ability of chemical agents to dislodge bacterial biofilm, the first step was to find the most optimal concentration and time of exposure dislodging the highest amount of the bacteria from the surface. The biofilms of *S. epidermidis*, *S. aureus*, *E. coli* and *P. aeruginosa* were treated at different concentrations and time points. The concentrations 25 mM EDTA and 1 g/L DTT were chosen as they showed significant increase in CFU count at 15 min compared to other time points when *P. aeruginosa* and *S. epidermidis* biofilms were investigated. Concentration of 1 g/L DTT was also proposed by other authors [14]. We did not observe any difference in CFU count when these concentrations were applied at different time points, therefore the time of 15 min was chosen as a most appropriate time for the routine microbiological examination.

Significantly higher CFU counts of *S. epidermidis*, *S. aureus*, *E. coli* and *P. aeruginosa* biofilm were detected after sonication compared to chemical dislodgement methods. The

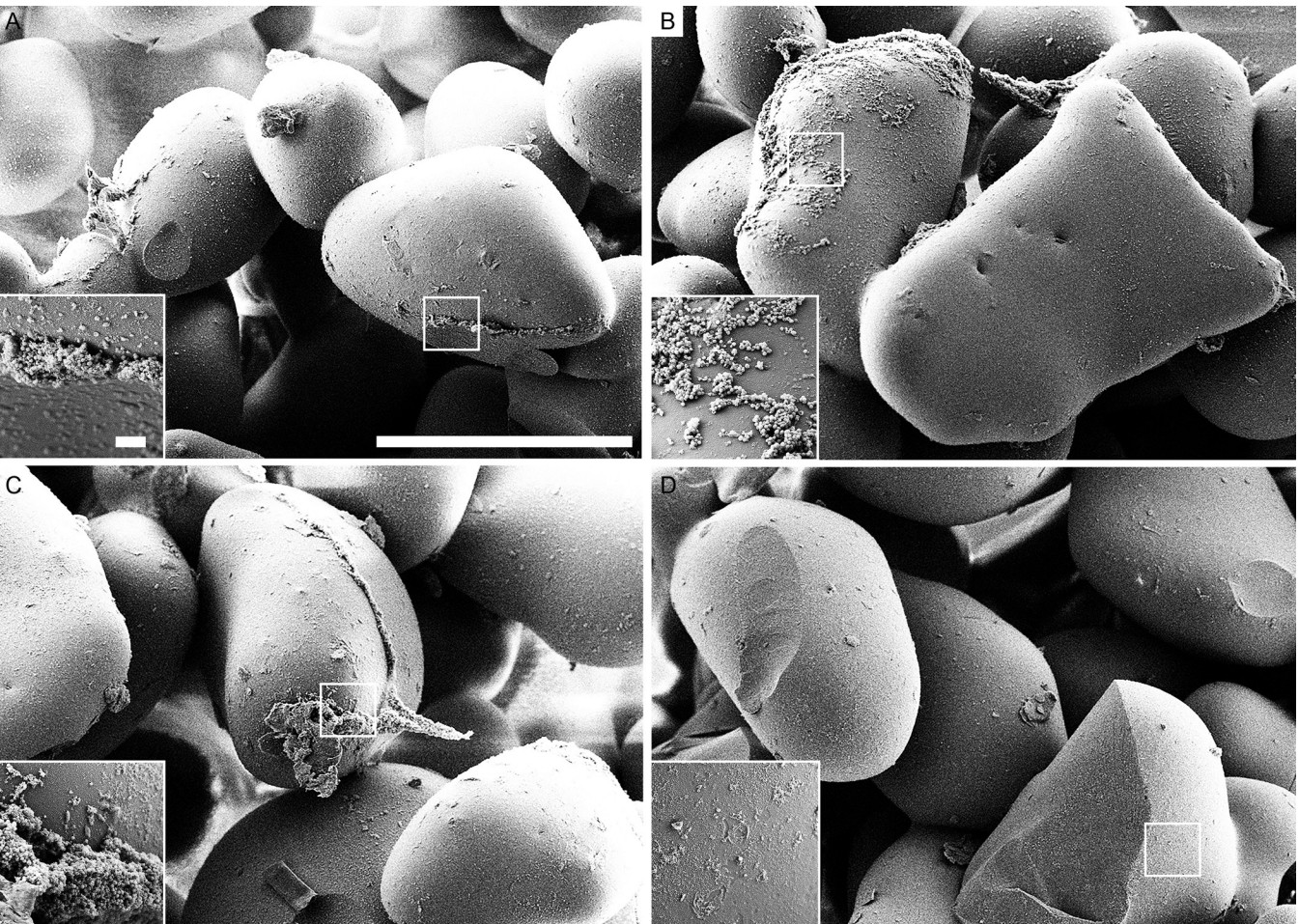

**Fig 4. Scanning electron microscopy (SEM) of *S. epidermidis* biofilm.** (A) beads after 0.9% NaCl treatment (control). (B) beads after EDTA treatment. (C) beads after DTT treatment. (D) beads after sonication treatment. Scale bars: 200 μm (inserts in the images represent 5 μm).

concentrations of EDTA (25 mM) and DTT (1 g/L) and sonication showed no impact on bacterial growth (S1 Fig).

Interestingly, our findings contradict the previously published results [14]. In their study, the authors investigated in vitro the dislodgement effect of DTT on polyethylene and titanium discs colonized with *S. aureus*, *S. epidermidis*, *P. aeruginosa* and *E. coli* biofilms. The authors found that DTT at 1 g/L applied for 15 min dislodged *P. aeruginosa* and *E. coli* biofilms with similar yield as with sonication, whereas the dislodgement of *S. aureus* and *S epidermidis* biofilms was even more efficient than with sonication.

Recently published *ex vivo* studies showed that treatment of explanted prosthesis with DTT may be superior to sonication for the diagnosis of periprosthetic joint infection [25–28]. The different type of biomaterial used for *ex vivo* biofilm studying may in part explain the discordance of results.

Similar discrepancy was found with EDTA. In our study, EDTA was unable to dislodge bacterial biofilms and the colony counts were similar to those obtained after treatment with 0.9% NaCl and were significantly lower compared to sonication. In contrast, previous authors demonstrated that EDTA affects *P. aeruginosa* biofilms [13, 29]. Banin et al. observed that exposure of *P. aeruginosa* biofilms to 50 mM EDTA dislodged biofilms.

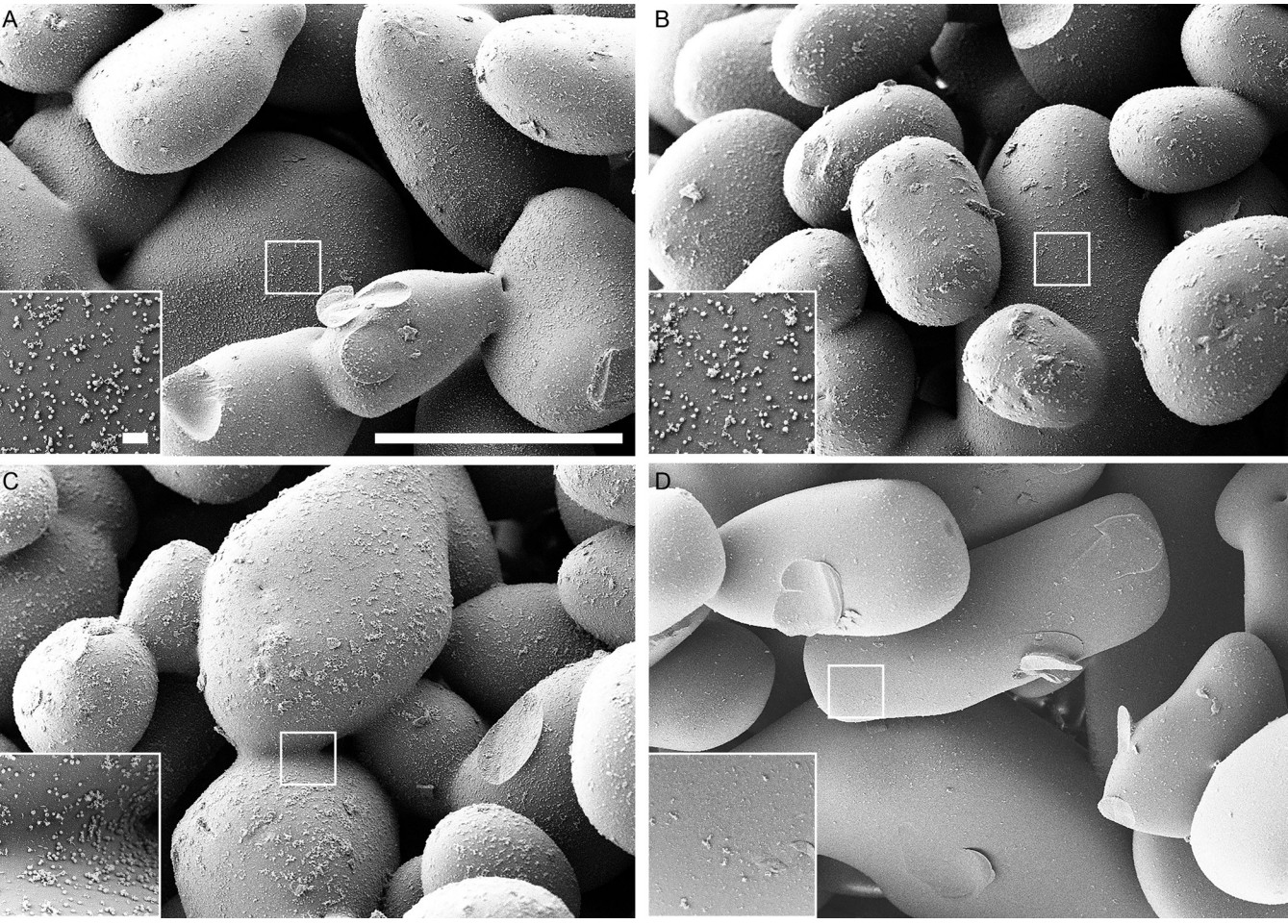

**Fig 5. Scanning electron microscopy (SEM) of *S. aureus* biofilm.** (A) beads after 0.9% NaCl treatment (control). (B) beads after EDTA treatment. (C) beads after DTT treatment. (D) beads after sonication treatment. Scale bars: 200 μm (inserts in the images represent 5 μm).

Addition of EDTA to the medium reservoir in a flow system increased the number of dislodged bacteria by >2 $\log_{10}$ CFU/mL after 50 min-incubation in the effluent compared to untreated flow system. The authors also showed that the activity of EDTA in biofilm detachment is mediated by chelation of several divalent cations such as magnesium, calcium, and iron that are required to stabilize the biofilm matrix. Our results derived from colony counting of dislodged bacterial cells were confirmed by two additional independent techniques, namely isothermal microcalorimetry and SEM imaging. Isothermal microcalorimetry is a highly sensitive method that enables a real-time monitoring of bacterial viability in terms of metabolism-related heat production. This method was widely used and validated for testing the anti-biofilm activity [20, 21, 30–34]. Here it was used to evaluate bacteria remaining on the glass beads after dislodging treatments. Isothermal microcalorimetry showed a significant delay in the detection of bacterial metabolism-related heat production from the beads with *S. epidermidis, S. aureus. E. coli and P aeruginosa*, when sonication was applied, as compared to chemical treatments—EDTA and DTT. These findings suggest that significantly less bacteria remained attached to the beads after sonication.

To visualize the bacteria remaining in the biofilms on the glass beads surface after treatment with either chemicals or sonication methods, the SEM was used. SEM micrographs have a

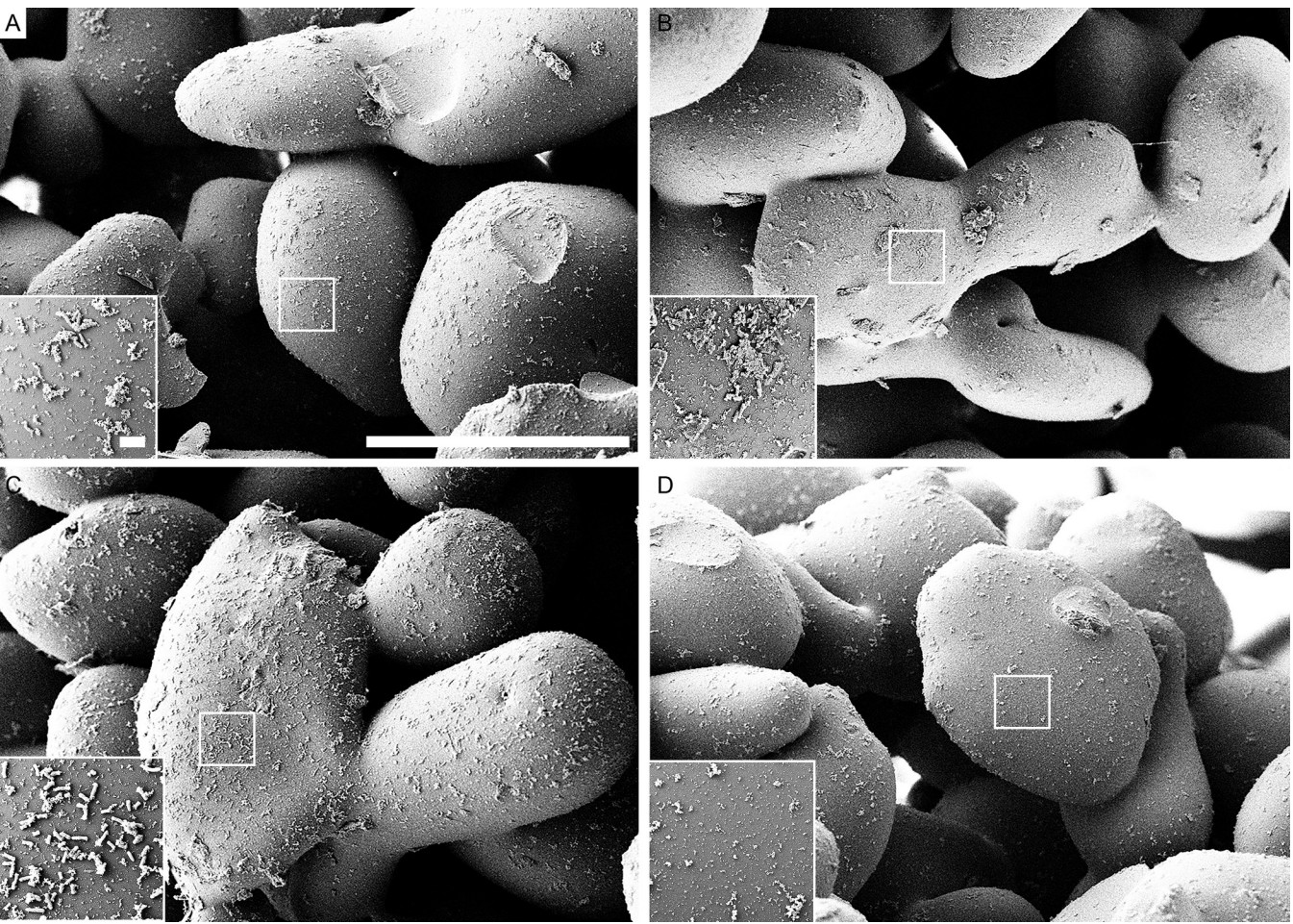

**Fig 6. Scanning electron microscopy (SEM) of *E. coli* biofilm.** (A) beads after 0.9% NaCl treatment (control). (B) beads after EDTA treatment. (C) beads after DTT treatment. (D) beads after sonication treatment. Scale bars: 200 μm (inserts in the images represent 5 μm).

large depth of field yielding a three-dimensional appearance, which is useful for understanding the surface structure of the sample. This method has been employed in various other studies providing good information on spatial structure [35, 36]. In our study, all types of bacterial biofilm SEM images showed less biofilm remaining on the beads when sonication was applied compared to the untreated control as well as both chemical methods.

There are several limitations of this study. First, anaerobes (e.g. *Cutibacterium* spp.) were not tested. Despite chemical methods were inferior to sonication in our study with all tested microorganisms (*S. epidermidis*, *S. aureus E. coli* and *P. aeruginosa*) it is possible that anaerobes may show better results with chemical methods due to their lower susceptibility to sonication. Before any recommendations about the clinical use in the routine microbiology testing, additional pathogens should be investigated. Second, for biofilm formation we used only laboratory strains. Typically clinical strains show larger variability therefore to evaluate a new diagnostic method *in vitro* the laboratory strains are more suitable. Third, we used only porous glass beads for biofilm formation. The porous glass beads possess a high volume-to-surface ratio therefore this model to form bacterial biofilm is probably more suitable for biofilm investigation than smooth materials. These results derived from this *in vitro* analysis represent a fundament for further exploration in the clinical setting with clinical

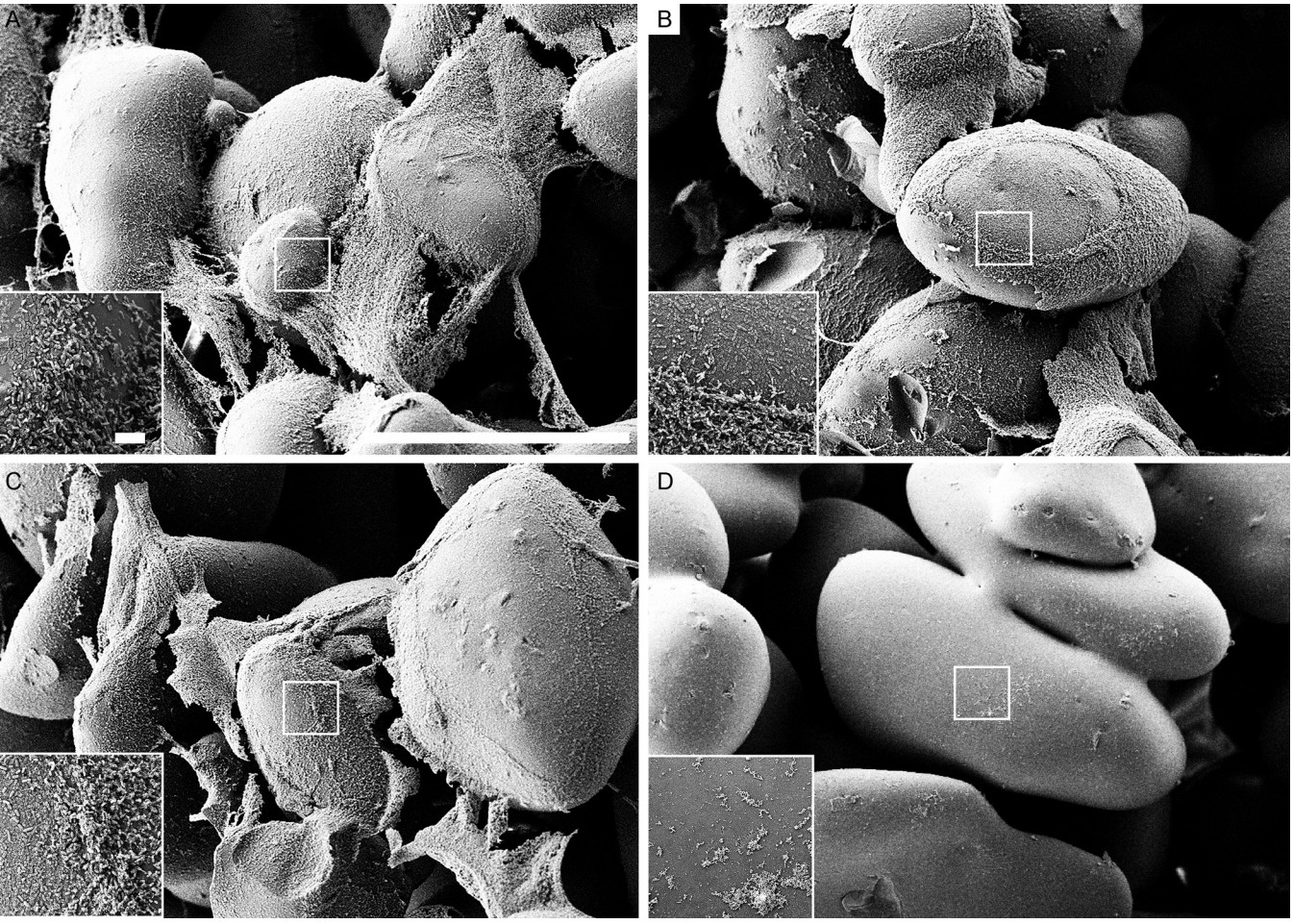

**Fig 7. Scanning electron microscopy (SEM) of *P. aeruginosa* biofilm.** (A) beads after 0.9% NaCl treatment (control). (B) beads after EDTA treatment. (C) beads after DTT treatment. (D) beads after sonication treatment. Scale bars: 200 μm (inserts in the images represent 5 μm).

strains and real implants. Forth, we incubated glass beads in the bacterial inoculum for 3 days until a visible biofilm was formed as described previously [14]. We assumed that further cultivation of mature biofilm to compare the ability of different methods for biofilm dislodgement is not needed. However it remains unknown, whether the ability of sonication or chemical methods for biofilm dislodgement would be different in more mature biofilms for example in the clinical setting when we deal with chronic implant-associated infections. Fifth, to study biofilm on glass beads surface, two complementary methods were used for detection of the remaining biofilms (microcalorimetry and scanning electron microscopy). Recently, novel methods for quantitative and qualitative evaluation of biofilm formation were evaluated. The BioTimer assay enumerates adherent microorganisms through microbial metabolism [37]. It showed promising results in the diagnosis of implant-associated infections, especially in combination with sonication, representing a simple and accurate way for the identification and enumeration of microorganisms. Other methods include confocal laser scanning microscopy (CLSM) [38], fluorescence microscopy [39] and atomic force microscopy [40] were not performed since both independent methods used in our study (microcalorimetry and scanning electron microscopy) correlated well.

## Conclusions

We showed that sonication is superior to the chemical method for dislodgement of bacterial biofilms of *S. epidermidis*, *S. aureus*, *E. coli* and *P. aeruginosa* from artificial surface. Therefore, sonication remains the primary assay for biofilm detection in the microbiological diagnosis of implant-associated infection. Future studies may investigate a potential additive effect of chemical dislodgement to sonication.

## Supporting information

**S1 Fig. The viability of planktonic bacteria in presence of chemical agents and sonication.**
(TIF)

## Acknowledgments

We thank the Core Facility for Electron Microscopy of the Charité –Universitätsmedizin Berlin for support in acquisition and analysis of the data. We also thank Sabine Bartosch from Berlin-Brandenburg School for Regenerative Therapies (BSRT), Berlin for careful review of the manuscript and useful comments.

## Author Contributions

**Conceptualization:** Svetlana Karbysheva, Mariagrazia Di Luca, Tobias Winkler, Michael Schütz, Andrej Trampuz.

**Data curation:** Svetlana Karbysheva, Mariagrazia Di Luca, Maria Eugenia Butini, Tobias Winkler, Michael Schütz, Andrej Trampuz.

**Formal analysis:** Svetlana Karbysheva, Mariagrazia Di Luca, Maria Eugenia Butini, Andrej Trampuz.

**Investigation:** Svetlana Karbysheva, Mariagrazia Di Luca, Maria Eugenia Butini.

**Methodology:** Svetlana Karbysheva, Mariagrazia Di Luca, Tobias Winkler, Michael Schütz, Andrej Trampuz.

**Project administration:** Svetlana Karbysheva, Tobias Winkler, Michael Schütz, Andrej Trampuz.

**Software:** Svetlana Karbysheva.

**Supervision:** Tobias Winkler, Michael Schütz, Andrej Trampuz.

**Validation:** Svetlana Karbysheva, Mariagrazia Di Luca, Michael Schütz, Andrej Trampuz.

**Visualization:** Svetlana Karbysheva, Tobias Winkler, Andrej Trampuz.

**Writing – original draft:** Svetlana Karbysheva, Mariagrazia Di Luca, Maria Eugenia Butini.

**Writing – review & editing:** Tobias Winkler, Michael Schütz, Andrej Trampuz.

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
