## [Decision Letter · Decision Letter 0]

20 Nov 2019

PONE-D-19-27719

Comparison of sonication with chemical biofilm dislodgement methods using chelating and reducing agents: potential implications for the microbiological diagnosis of implant associated infection

PLOS ONE

Dear Prof. Trampuz,

Thank you for submitting your manuscript to PLOS ONE. After careful consideration, we feel that it has merit but does not fully meet PLOS ONE’s publication criteria as it currently stands. Therefore, we invite you to submit a revised version of the manuscript that addresses the points raised during the review process.

We would appreciate receiving your revised manuscript by Jan 04 2020 11:59PM. To enhance the reproducibility of your results, we recommend that if applicable you deposit your laboratory protocols in protocols.io, where a protocol can be assigned its own identifier (DOI) such that it can be cited independently in the future. For instructions see: http://journals.plos.org/plosone/s/submission-guidelines#loc-laboratory-protocols

We look forward to receiving your revised manuscript.

Kind regards,

Olivier Habimana

Academic Editor

PLOS ONE

Journal Requirements:

Additional Editor Comments (if provided):

The manuscript has now been reviewed by two experts in the field, with one recommended major revisions, and the other rejection. Major concerns pertained to the clinical relevance of the study that also is in need of more experimentation. The authors are, therefore, kindly requested to present a rebuttal addressing these concerns for further deliberation of the manuscript.

Reviewers' comments:

Reviewer's Responses to Questions

**Comments to the Author**

1. Is the manuscript technically sound, and do the data support the conclusions?

Reviewer #1: Yes

Reviewer #2: Yes

2. Has the statistical analysis been performed appropriately and rigorously? 

Reviewer #1: Yes

Reviewer #2: Yes

3. Have the authors made all data underlying the findings in their manuscript fully available?

Reviewer #1: Yes

Reviewer #2: Yes

4. Is the manuscript presented in an intelligible fashion and written in standard English?

Reviewer #1: Yes

Reviewer #2: Yes

5. Review Comments to the Author

Reviewer #1: The authors analyzed the activity (not efficacy as this is an in-vitro study) of chemical methods to be used for biofilm removal.

The manuscript is well written; the laboratory methods are suitable. The biofilm dispersion was determined by different methods. However, for this is a preliminary study. The purpose of removing biofilms from implants is medical microbiological diagnostics. For that, more microorganisms (incl. clinical strains) should be included. EDTA may interfere with viability of microorganisms. The supplementary use of chemicals to sonication should included in that study.

Minor remarks. Microorganisms associated with implant infections should be introduced.

How do the glass beads for biofilm formation correspond to clinically used material?

Reviewer #2: The authors performed an interesting in vitro study with the aim to evaluate the diagnostic performance of EDTA and DTT agents in comparison with the reference method (sonication) for bacterial dislodgement of S epidermidis and P aeruginosa biofilms.

Although the study has the following limitations:

- only 2 strains were tested (lacking information on S aureus, which is a well known biofilm producer),

- only reference strains were used,

- only glass beads as foreign body materials,

it undoubtely confirmed the higher bacterial detection rate of sonication method compared with other techniques. cThis finding has important consequences in terms of the clinical application of these new techniques, which might be considered as complentary, and not alternative, to sonication.

Detailed comment:

- Why 3 days of biofilm formation? Would the authors believe the results would be different with shorter or longer incubation time? Please comment on it

- The authors should also stress the potential use of sonication in implant infections other than orthopedics (ie intracardiac devices; neurosurgical shunts). Please add some references and a comment on it in the discussion.

- Insert a reference or an explanation on the use of porous glass beads as suitable model for biofilm formation in vitro in the method part.

- Please add a reference or an explanation for the timing of EDTA and DTT exposition for biofilm dislodgement in the method part, not only in the discussion.

- Furthermore, which is the rationale of choosing the EDTA and DTT concentrations? Please explain also in the method part.

- Insert the limit of detection in the method part (expressed as CFU/mL)

- Were the sonicated beads also submitted to vortexing process?

- The viability experiment should be better explained or addressed in an appropriate reference.

- When comparing all the three quantitative different groups, was the ANOVA test performed? In the case the ANOVA test was not used, please indicate the reason.

- Figure1 and Figure2; Figure5, 6. Please write per extenso EDTA and DTT. NS should be explained.

- Figure3 and figure4. Insert the different methods in the legend.

- Figure4. TTD should be explained. Which is the meaning of the arrow? Please explain.

- Figure3. 0,9% or 0.9%?. Please correct.

- Lines 151-153: the authors should describe that these CFUs obtained with these concentrations and timing derive from the above described experiments.

- Line 155: insert CFU/mL

- Lines 156-157: at which concentration and timing of EDTA and DTT for P. aeruginosa? Please describe.

- Lines 158-160: this sentence might be moved in a more appropriate place (ie, at the beginning of the paragraph).

- Lines 173-177: the results of P aeruginosa are similar to those observed for S epidermidis. Please re-phrase the sentence and remove the term “although”, which might be confounding for the reader.

- Lines 225-226. This sentence is not necessary since it has been already stated.

- It would be interesting to have a comment from the authors regarding other non-culture based methods of diagnosis of implant associated infection (ie, the metabolic assay BTA, BioTimer Assay, which has been recently investigated in intracardiac and central venous catheters infections as a complementary method to sonication)

- Please include a sentence highlighting the limitations of the study

6. PLOS authors have the option to publish the peer review history of their article (what does this mean?). If published, this will include your full peer review and any attached files.

Reviewer #1: No

Reviewer #2: No

---

## [Author Response · Author response to Decision Letter 0]

12 Feb 2020

Karbysheva et al. Manuscript PONE-D-19-27719

Comparison of sonication with chemical biofilm dislodgement methods using chelating and reducing agents: implications for the microbiological diagnosis of implant associated infection

Dear Editor

Thank you for the opportunity to re-submit our scientific work again to PLOS ONE for a potential review. Please find below our response to the comments of the Reviewers. Based on these comments, we have already revised our manuscript and hope that with this improved quality it may be suitable to send it for external review and potentially of interest for your prestigious Journal. 

With best regards,

Andrej Trampuz

Point-by-point rebuttal

Scientific Editor Comments:

Reviewer #1: 

1. The authors analyzed the activity (not efficacy as this is an in-vitro study) of chemical methods to be used for biofilm removal.

COMMENT: We corrected in the manuscript.

“We compared the activity efficacy of chemical methods for biofilm dislodgement to the standard sonication procedure in an established in vitro model of artificial biofilm.”

2. The manuscript is well written; the laboratory methods are suitable. The biofilm dispersion was determined by different methods. However, for this is a preliminary study. The purpose of removing biofilms from implants is medical microbiological diagnostics. For that, more microorganisms (incl. clinical strains) should be included. 

COMMENT: Additionally we investigated two more microorganisms (S. aureus and E. coli).

3. EDTA may interfere with viability of microorganisms. The supplementary use of chemicals to sonication should included in that study.

COMMENT: We added supplementary material on the viability of planktonic bacteria in presence of chemical agents and sonication.

4. Minor remarks. Microorganisms associated with implant infections should be introduced.

COMMENT: We added in the manuscript additional information about microorganisms associated with implant infections.

“Most commonly isolated microorganisms in patients with periprosthetic joint infection are coagulase-negative staphylococci (30-45%) and Staphylococcus aureus (12-23%), followed by streptococci (9-10%), enterococci (3-7%), gram-negative bacilli (3-6%) and anaerobes (2-4%) [6]. Similar distribution of pathogens is observed in CIED [2] and neurosurgical shunt-associated infections [4].”

5. How do the glass beads for biofilm formation correspond to clinically used material?

COMMENT: We added the explanation and references in the method part. 

“As a model to form the bacterial biofilm porous glass beads (diameter 4 mm, pore sizes 60 μm, ROBU®, Hattert, Germany) were used. Due to the high volume-to-surface ratio, glass beads were used for biofilm studies rather than smooth materials, as investigated in numerous previous research works regarding biofilm formation and anti-biofilm activity [15-20].”

Reviewer #2: 

The authors performed an interesting in vitro study with the aim to evaluate the diagnostic performance of EDTA and DTT agents in comparison with the reference method (sonication) for bacterial dislodgement of S epidermidis and P aeruginosa biofilms.

Although the study has the following limitations:

1. only 2 strains were tested (lacking information on S aureus, which is a well known biofilm producer).

Reply: Additionally we investigated two more microorganisms (S. aureus and E. coli).

2. only reference strains were used,

COMMENT: Explained in the manuscript.

“We did not use clinical strains as they typically show larger variability and are not suitable for investigation of a new diagnostic method.”

3. only glass beads as foreign body materials,

COMMENT: We added the explanation and references in the method part. 

“As a model to form the bacterial biofilm porous glass beads (diameter 4 mm, pore sizes 60 μm, ROBU®, Hattert, Germany) were used. Due to the high volume-to-surface ratio, glass beads were used for biofilm studies rather than smooth materials, as investigated in numerous previous research works regarding biofilm formation and anti-biofilm activity [15-20].”

It undoubtely confirmed the higher bacterial detection rate of sonication method compared with other techniques. This finding has important consequences in terms of the clinical application of these new techniques, which might be considered as complentary, and not alternative, to sonication.

4. Why 3 days of biofilm formation? Would the authors believe the results would be different with shorter or longer incubation time? Please comment on it

COMMENT: We incubated glass beads for 3 days for biofilm formation as described previously by Drago et al. Clin Orthop Relat Res. 2012. The authors grew the biofilm for 3 days until a visible biofilm was formed therefor further cultivation of mature biofilm is not needed.

5. The authors should also stress the potential use of sonication in implant infections other than orthopedics (ie intracardiac devices; neurosurgical shunts). Please add some references and a comment on it in the discussion.

COMMENT: We added the addition information in the manuscript.

“Implants Orthopedic devices are increasingly used to improve the mobility (joint replacement and bone fixation devices) in the treatment of degenerative joint disease (osteoarthritis) and for the fixation of bone fractures or enhance the survival and assist the performance of physiological functions (cardiac implantable electronic device (CIED) and neurosurgical shunts). Infections represent a significant complication of implant surgery, resulting in major challenges regarding the diagnosis and treatment [1-5].”

6. Insert a reference or an explanation on the use of porous glass beads as suitable model for biofilm formation in vitro in the method part.

COMMENT: We added the explanation and references in the method part.

7. Please add a reference or an explanation for the timing of EDTA and DTT exposition for biofilm dislodgement in the method part, not only in the discussion.

COMMENT: We added the explanation and references for the timing of EDTA and DTT exposition in the method part.

8. Furthermore, which is the rationale of choosing the EDTA and DTT concentrations? Please explain also in the method part.

COMMENT: We added the explanation of choosing EDTA and DTT concentration in the method part.

9. Insert the limit of detection in the method part (expressed as CFU/mL)

COMMENT: We corrected in the manuscript as CFU/mL.

10. Were the sonicated beads also submitted to vortexing process?

COMMENT: We did use the vortexing process in sonication protocol. We corrected the protocol in the method part.

“Briefly, each bead was inoculated in 1 ml 0.9% saline, vortexed for 30 sec, and sonicated at 40 kHz at intensity 0.1 Watt/cm2 (BactoSonic, BANDELIN electronic, Berlin, Germany) for 1 min and vortexed again for 30 sec.”

11. The viability experiment should be better explained or addressed in an appropriate reference.

COMMENT: We added supplementary material on the viability of planktonic bacteria in presence of chemical agents and sonication.

12. When comparing all the three quantitative different groups, was the ANOVA test performed? In the case the ANOVA test was not used, please indicate the reason.

COMMENT: We recalculated the result of the study using ANOVA test. New results are added in the manuscript.

13. Figure1 and Figure2; Figure5, 6. Please write per extenso EDTA and DTT. NS should be explained.

COMMENT: We corrected the information in the Figures.

14. Figure3 and figure4. Insert the different methods in the legend.

COMMENT: We corrected the information in the Figures.

15. Figure4. TTD should be explained. Which is the meaning of the arrow? Please explain.

COMMENT: We corrected the information in the Figures.

16. Figure3. 0,9% or 0.9%?. Please correct.

COMMENT: We corrected the information in the Figures.

17. Lines 151-153: the authors should describe that these CFUs obtained with these concentrations and timing derive from the above described experiments.

COMMENT: We corrected the information in the method part.

18. Line 155: insert CFU/mL

COMMENT: We corrected in the manuscript as CFU/mL.

18. Lines 156-157: at which concentration and timing of EDTA and DTT for P. aeruginosa? Please describe.

COMMENT: We corrected the information in the results part.

19. Lines 158-160: this sentence might be moved in a more appropriate place (ie, at the beginning of the paragraph).

COMMENT: We corrected the information in the results part.

20. Lines 173-177: the results of P aeruginosa are similar to those observed for S epidermidis. Please re-phrase the sentence and remove the term “although”, which might be confounding for the reader.

COMMENT: We corrected the information in the results part.

21. Lines 225-226. This sentence is not necessary since it has been already stated.

COMMENT: We corrected the information in the results part.

22. It would be interesting to have a comment from the authors regarding other non-culture based methods of diagnosis of implant associated infection (ie, the metabolic assay BTA, BioTimer Assay, which has been recently investigated in intracardiac and central venous catheters infections as a complementary method to sonication)

- Please include a sentence highlighting the limitations of the study

COMMENT: We added the additional information on non-culture based methods of diagnosis of implant associated infection in the limitation of the study.

---

## [Decision Letter · Decision Letter 1]

17 Mar 2020

PONE-D-19-27719R1

Comparison of sonication with chemical biofilm dislodgement methods using chelating and reducing agents: implications for the microbiological diagnosis of implant associated infection

PLOS ONE

Dear Prof. Trampuz,

Thank you for submitting your manuscript to PLOS ONE. After careful consideration, we feel that it has merit but does not fully meet PLOS ONE’s publication criteria as it currently stands. Therefore, we invite you to submit a revised version of the manuscript that addresses the points raised during the review process.

We would appreciate receiving your revised manuscript by May 01 2020 11:59PM. To enhance the reproducibility of your results, we recommend that if applicable you deposit your laboratory protocols in protocols.io, where a protocol can be assigned its own identifier (DOI) such that it can be cited independently in the future. For instructions see: http://journals.plos.org/plosone/s/submission-guidelines#loc-laboratory-protocols

We look forward to receiving your revised manuscript.

Kind regards,

Olivier Habimana

Academic Editor

PLOS ONE

Reviewers' comments:

Reviewer's Responses to Questions

**Comments to the Author**

1. If the authors have adequately addressed your comments raised in a previous round of review and you feel that this manuscript is now acceptable for publication, you may indicate that here to bypass the “Comments to the Author” section, enter your conflict of interest statement in the “Confidential to Editor” section, and submit your "Accept" recommendation.

Reviewer #2: All comments have been addressed

Reviewer #3: (No Response)

2. Is the manuscript technically sound, and do the data support the conclusions?

Reviewer #2: Yes

Reviewer #3: Yes

3. Has the statistical analysis been performed appropriately and rigorously? 

Reviewer #2: Yes

Reviewer #3: Yes

4. Have the authors made all data underlying the findings in their manuscript fully available?

Reviewer #2: Yes

Reviewer #3: Yes

5. Is the manuscript presented in an intelligible fashion and written in standard English?

Reviewer #2: Yes

Reviewer #3: Yes

6. Review Comments to the Author

Reviewer #2: (No Response)

Reviewer #3: Thank you for letting me reviewing the revision of this manuscript. I’m impressed by the extensive and multimodal analysis of the activity of different biofilm-dislodging methods by the authors. The results are certainly of clinical relevance.

The former reviewers comments were addressed appropriately with a few exceptions:

Reviewer #2

2. only reference strains were used,

COMMENT: Explained in the manuscript.

“We did not use clinical strains as they typically show larger variability and are not suitable for

investigation of a new diagnostic method.”

3. only glass beads as foreign body materials,

COMMENT: We added the explanation and references in the method part.

“As a model to form the bacterial biofilm porous glass beads (diameter 4 mm, pore sizes 60 μm,

ROBU®, Hattert, Germany) were used. Due to the high volume-to-surface ratio, glass beads

were used for biofilm studies rather than smooth materials, as investigated in numerous

previous research works regarding biofilm formation and anti-biofilm activity [15-20].”

It undoubtedly confirmed the higher bacterial detection rate of sonication method compared with

other techniques. This finding has important consequences in terms of the clinical application of

these new techniques, which might be considered as complementary, and not alternative, to

sonication.

• These two limitations should be stated in the discussion. This in vitro analysis represents a fundament for further exploration in the clinical setting with clinical strains and real implants.

4. Why 3 days of biofilm formation? Would the authors believe the results would be different

with shorter or longer incubation time? Please comment on it

COMMENT: We incubated glass beads for 3 days for biofilm formation as described previously

by Drago et al. Clin Orthop Relat Res. 2012. The authors grew the biofilm for 3 days until a

visible biofilm was formed therefor further cultivation of mature biofilm is not needed.

• I agree with the reviewer. It remains unknown, whether the results would be different in more mature biofilms (as we are dealing with in the clinical setting). Comment this in the limitations setting.

9. Insert the limit of detection in the method part (expressed as CFU/mL)

COMMENT: We corrected in the manuscript as CFU/mL.

• Could not find the detection limit in the methods part. Please make sure to provide the limit of detection.

14. Figure3 and figure4 . Insert the different methods in the legend.

COMMENT: We corrected the information in the Figures.

• Please mention the different methods tested in the legend.

15. Figure4. TTD should be explained. Which is the meaning of the arrow? Please explain.

COMMENT: We corrected the information in the Figures.

• Please explain the meaning of the arrow in the legend

• Please make sure, that the figures are numbered correctly and mentioned accordingly in the manuscript text. Figure 2 is now Fig 5, Fig 3 is now Fig 6, etc.

18. Line 155: insert CFU/mL

COMMENT: We corrected in the manuscript as CFU/mL.

• Please add „CFU/ml“ in al the comparisons in the entire section (e.g. lines 170, 173, 178, 181)

Additional comments from my side:

• Make sure to use the terms dislodgement (vs. dislodgment) and NaCl (vs. saline) uniformally throughout the manuscript

• L 142: Please provide an introducing sentence before reporting the results. For example: „Figure 1 shows CFU counts of residual biofilms after exposure to chemical agents for variable durations.“

7. PLOS authors have the option to publish the peer review history of their article (what does this mean?). If published, this will include your full peer review and any attached files.

Reviewer #2: No

Reviewer #3: No

---

## [Author Response · Author response to Decision Letter 1]

20 Mar 2020

Karbysheva et al. Manuscript PONE-D-19-27719

Comparison of sonication with chemical biofilm dislodgement methods using chelating and reducing agents: implications for the microbiological diagnosis of implant associated infection

Dear Editor

Thank you for the opportunity to re-submit our scientific work again to PLOS ONE for a potential review. Please find below our response to the comments of the Reviewers. Based on these comments, we have already revised our manuscript and hope that with this improved quality it may be suitable to send it for external review and potentially of interest for your prestigious Journal. 

With best regards,

Andrej Trampuz

Point-by-point rebuttal

Scientific Editor Comments:

Reviewer #2

First revision: 

2. only reference strains were used,

COMMENT: Explained in the manuscript.

“We did not use clinical strains as they typically show larger variability and are not suitable for

investigation of a new diagnostic method.”; 

3. only glass beads as foreign body materials,

COMMENT: We added the explanation and references in the method part.

“As a model to form the bacterial biofilm porous glass beads (diameter 4 mm, pore sizes 60 μm,

ROBU®, Hattert, Germany) were used. Due to the high volume-to-surface ratio, glass beads

were used for biofilm studies rather than smooth materials, as investigated in numerous

previous research works regarding biofilm formation and anti-biofilm activity [15-20].”

Second revision, Reviewer #2: It undoubtedly confirmed the higher bacterial detection rate of sonication method compared with other techniques. This finding has important consequences in terms of the clinical application of these new techniques, which might be considered as complementary, and not alternative, to sonication.

These two limitations should be stated in the discussion. This in vitro analysis represents a fundament for further exploration in the clinical setting with clinical strains and real implants.

COMMENT: We stated these two limitations in the discussion.

“.., for biofilm formation we used only laboratory strains. Typically clinical strains show larger variability therefore to evaluate a new diagnostic method in vitro the laboratory strains are more suitable. Third, we used only porous glass beads for biofilm formation. The porous glass beads possess a high volume-to-surface ratio therefore this model to form bacterial biofilm is probably more suitable for in vitro biofilm investigation than smooth materials. The results derived from this in vitro analysis represent a fundament for further exploration in the clinical setting with clinical strains and real implants.”

First revision: 

4. Why 3 days of biofilm formation? Would the authors believe the results would be different

with shorter or longer incubation time? Please comment on it

COMMENT: We incubated glass beads for 3 days for biofilm formation as described previously

by Drago et al. Clin Orthop Relat Res. 2012. The authors grew the biofilm for 3 days until a

visible biofilm was formed therefor further cultivation of mature biofilm is not needed.

Second revision, Reviewer #2: I agree with the reviewer. It remains unknown, whether the results would be different in more mature biofilms (as we are dealing with in the clinical setting). Comment this in the limitations setting.

COMMENT: We stated this limitation in the discussion.

“Forth, we incubated glass beads in the bacterial inoculum for 3 days until a visible biofilm was formed as described previously [14]. We assumed that further cultivation of mature biofilm to compare the ability of different methods for biofilm dislodgement is not needed. However it remains unknown, whether the ability of sonication or chemical methods for biofilm dislodgement would be different in more mature biofilms for example in the clinical setting when we deal with chronic implant-associated infections.”

First revision: 

9. Insert the limit of detection in the method part (expressed as CFU/mL)

COMMENT: We corrected in the manuscript as CFU/mL.

Second revision, Reviewer #2: Could not find the detection limit in the methods part. Please make sure to provide the limit of detection.

COMMENT: We insert the limit of detection in the method part.

“The serial dilutions allowed to raise the upper limit of detection providing a reportable range from 0 to 100,000,000 CFU/mL.”

First revision: 

14. Figure3 and figure4 . Insert the different methods in the legend.

COMMENT: We corrected the information in the Figures.

Second revision, Reviewer #2: Please mention the different methods tested in the legend.

COMMENT: Figure 3 is now Figure 2. We insert the methods in the legend.

First revision: 

15. Figure4. TTD should be explained. Which is the meaning of the arrow? Please explain.

COMMENT: We corrected the information in the Figures.

Second revision, Reviewer #2: Please explain the meaning of the arrow in the legend

COMMENT: Figure 4 is now Figure 3. We explained the meaning of TTD in the legend.

Second revision, Reviewer #2: Please make sure, that the figures are numbered correctly and mentioned accordingly in the manuscript text. Figure 2 is now Fig 5, Fig 3 is now Fig 6, etc.

COMMENT: We revised the numbers of the figures and mentioned them correctly in the manuscript. We combined Figure 1 and Figure 2 in Figure 1. Figure 3 is now Figure 2. Figure 4 is now Figure 3. Figure 5 is now Figure 4. Figure 6 is now Figure 7. We added two additional Figures (5 and 6).

First revision: 18. Line 155: insert CFU/mL

COMMENT: We corrected in the manuscript as CFU/mL.

Second revision, Reviewer #2: Please add „CFU/ml“ in al the comparisons in the entire section (e.g. lines 170, 173, 178, 181)

COMMENT: We added “CFU/mL” in the manuscript.

Additional comments from Reviewer #2:

• Make sure to use the terms dislodgement (vs. dislodgment) and NaCl (vs. saline) uniformally throughout the manuscript

COMMENT: We corrected the terms in the manuscript.

• L 142: Please provide an introducing sentence before reporting the results. For example: „Figure 1 shows CFU counts of residual biofilms after exposure to chemical agents for variable durations.“

COMMENT: We added the sentence in the results part.

---

## [Editor Report · Decision Letter 2]

24 Mar 2020

Comparison of sonication with chemical biofilm dislodgement methods using chelating and reducing agents: implications for the microbiological diagnosis of implant associated infection

PONE-D-19-27719R2

Dear Dr. Trampuz,

We are pleased to inform you that your manuscript has been judged scientifically suitable for publication and will be formally accepted for publication once it complies with all outstanding technical requirements.

With kind regards,

Olivier Habimana

Academic Editor

PLOS ONE
---

## [Editor Report · Acceptance letter]

26 Mar 2020

PONE-D-19-27719R2 

Comparison of sonication with chemical biofilm dislodgement methods using chelating and reducing agents: implications for the microbiological diagnosis of implant associated infection 

Dear Dr. Trampuz:

I am pleased to inform you that your manuscript has been deemed suitable for publication in PLOS ONE. Congratulations! Your manuscript is now with our production department. 

With kind regards,

on behalf of

Dr. Olivier Habimana 

Academic Editor

PLOS ONE